# Ethanol Coupling Reactions over MgO–Al_2_O_3_ Mixed Oxide-Based Catalysts for Producing Biofuel Additives

**DOI:** 10.3390/molecules28093788

**Published:** 2023-04-28

**Authors:** Anna Vikár, Ferenc Lónyi, Amosi Makoye, Tibor Nagy, Gyula Novodárszki, Róbert Barthos, Blanka Szabó, József Valyon, Magdolna R. Mihályi, Dhanapati Deka, Hanna E. Solt

**Affiliations:** 1Institute of Materials and Environmental Chemistry, Research Centre for Natural Sciences, Magyar Tudósok Körútja 2, 1117 Budapest, Hungary; 2Hevesy György Doctoral School of Chemistry, ELTE Eötvös Loránd University, Pázmány Péter s. 1/A, 1117 Budapest, Hungary; 3Biomass Conversion Laboratory, Department of Energy, Tezpur University, Tezpur 784028, India

**Keywords:** ethanol coupling to butanol, Guerbet reaction, MgO–Al_2_O_3_ mixed oxide-based catalyst, DRIFT spectroscopy

## Abstract

Catalytic conversion of ethanol to 1-butanol was studied over MgO–Al_2_O_3_ mixed oxide-based catalysts. Relationships between acid-base and catalytic properties and the effect of active metal on the hydrogen transfer reaction steps were investigated. The acid-base properties were studied by temperature-programmed desorption of CO_2_ and NH_3_ and by the FT-IR spectroscopic examination of adsorbed pyridine. Dispersion of the metal promoter (Pd, Pt, Ru, Ni) was determined by CO pulse chemisorption. The ethanol coupling reaction was studied using a flow-through microreactor system, He or H_2_ carrier gas, WHSV = 1 $g_{EtOH}\cdot g_{cat.}^{-1}\cdot h^{-1}$, at 21 bar, and 200–350 °C. Formation and transformation of surface species under catalytic conditions were studied by DRIFT spectroscopy. The highest butanol selectivity and yield was observed when the MgO–Al_2_O_3_ catalyst contained a relatively high amount of strong-base and medium-strong Lewis acid sites. The presence of metal improved the activity both in He and H_2_; however, the butanol selectivity significantly decreased at temperatures ≥ 300 °C due to acceleration of undesired side reactions. DRIFT spectroscopic results showed that the active metal promoted H-transfer from H_2_ over the narrow temperature range of 200–250 °C, where the equilibrium allowed significant concentrations of both dehydrogenated and hydrogenated products.

## 1. Introduction

Conversion of bioethanol to biobutanol via the Guerbet coupling reaction attracted significant interest in the chemical industry, because 1-butanol could be used as a renewable fuel or a blending component of gasoline as well as the precursor for the production of other valuable chemicals [1,2,3].

According to the most accepted mechanism, the coupling reaction involves four consecutive reaction steps [2,3,4,5]: alcohol dehydrogenation to aldehyde, aldol addition, dehydration, and hydrogenation of the obtained unsaturated aldehyde to butanol. The aldol coupling takes place on basic centers presumably via surface enolate formation, whereas acidic centers are very active in dehydration. In the absence of an active metal center, the dehydrogenation of alcohol to aldehyde and hydrogenation of the unsaturated aldehyde intermediate can occur at the acid-base pair sites of the surface [2,3,4,5,6]. The overall ethanol coupling reaction does not require the presence of a dehydrogenating–hydrogenating metal component and/or hydrogen. The chemical state and binding form of hydrogen obtained in the first dehydrogenation step or its retake in the closing reaction steps leading to butanol formation are not fully understood. Several studies suggest that the hydrogenation of the aldehyde intermediate takes place mainly by Meerwein–Ponndorf–Verley (MPV) reduction with the reactant ethanol [2,3,4,6,7,8]. According to a proposed mechanism, first the C=O bond is reduced, then the obtained unsaturated alcohol is converted to an aldehyde by isomerization and tautomerization. Then, the aldehyde is reduced again by ethanol in the MPV reaction [8]. In the MPV reduction the ethanol transforms to acetaldehyde, which can take part in the butanol-generating chain process as shown in Figure 1 [2,9,10].

Some former studies also suggested that in the presence of some heterogeneous catalysts, another reaction path could open at higher temperatures in which two ethanol molecules were directly coupled [5,9,11]. According to this mechanism, one of the C–H bonds in the –CH_3_ group of ethanol is activated and reacts with the OH group of a second ethanol molecule, leading to formation of butanol and water molecules. It should be noted, however, that the mechanism of direct ethanol coupling was suggested for basic catalysts (MgO, hydroxyapatite, alkaline metal zeolites) in the absence of a metal component at temperatures 350–450 °C [2]. At lower reaction temperatures and/or in the presence of an active metal catalyst component, generally the aldol condensation pathway was considered to be the main reaction mechanism [12].

An active metal component having hydrogenation–dehydrogenation activity can promote hydrogen transfer reactions in the initial step of ethanol dehydrogenation and in the hydrogenation steps of the unsaturated aldehyde intermediate (Figure 1) and thereby can affect product distribution [2].

Depending on the catalyst and the applied reaction parameters, the butanol formation reaction is often accompanied by several undesired side reactions such as formation of ethylene, diethyl ether, different unsaturated and saturated ketones, hydroxy ketones, keto esters, and CO or CO_2_ from deoxygenation reactions [2,5,13]. Therefore, the selective catalyst must contain well-balanced catalytic functions to minimize possible side reactions. Mixed oxide-based MgO–Al_2_O_3_ catalysts are among the most often applied catalysts in the Guerbet coupling of alcohols, because tuning the Mg/Al ratio and the calcination temperature of the precursor hydrotalcite creates the possibility of changing the surface concentration of acid and base sites of different strength, thereby achieving the appropriate balance of aldol coupling and dehydration activities [5,13,14,15,16,17,18,19,20,21,22].

In the present study, the ethanol coupling reaction to butanol was investigated in a fixed-bed flow-through microreactor system over MgO–Al_2_O_3_ mixed oxide catalysts prepared from hydrotalcite by applying different calcination temperatures to reveal structure–activity relationships between acid-base and catalytic properties. Pd, Pt, Ni, and Ru promoters were introduced to see how the presence of dehydrogenating–hydrogenating function contributes to or modifies the hydrogen transfer reaction steps and, as a consequence, the catalytic activity and product selectivity.

## 2. Results

### 2.1. Catalyst Structure

X-ray powder diffraction (XRPD) patterns of metal-free and metal-containing MgO–Al_2_O_3_ mixed oxide samples, obtained by calcination of their hydrotalcite precursor at 550 °C, are shown in Figure 1. The diffractogram of the precursor material (Figure 1, a) confirms the formation of hydrotalcite structure during its synthesis. Calcination resulted in the transformation of hydrotalcite into MgO–Al_2_O_3_ mixed oxide, which shows only the characteristic reflections of the cubic periclase form of magnesium oxide (Figure 1, b). No crystalline aluminum oxide phase could be detected, indicating that Al is finely dispersed in the MgO structure, possibly resulting in the formation of Al–O–Mg linkages. The metal-containing samples exhibit patterns very similar to those of the metal-free MgO–Al_2_O_3_ samples (Figure 1, c–f). Characteristic reflections of metal oxide or metal phases are discernible only for the 1%Ru/MgO–Al_2_O_3_ sample (Figure 1, f), suggesting that their particle size is below the XRPD detection limit of ca. 5 nm in the catalysts containing Pd, Pt, or Ni.

The metal dispersion (*D*_Me_), that is, the ratio of surface metal atoms to the total number of metal atoms of the metal particles, was determined by the CO pulse chemisorption method. The *D*_Me_ calculation was carried out assuming an adsorbed CO to surface metal atom ratio of 1 to 1 for Pt, Ru, and Ni, whereas there was a stoichiometry of 1 to 2 for Pd (latter stoichiometry was suggested by the abundancy of bridge-bonded CO species on Pd) [23,24]. It is important to note that unlike the noble metal-containing samples, the temperature-programmed H_2_ reduction of 5%Ni/MgO–Al_2_O_3_ (not shown) indicated that only 5.2% of Ni could be reduced to a metallic state up to 600 °C, most probably due to the extremely strong interaction of nickel oxide and the alumina support [25]. Only the reduced fraction of Ni was considered in the calculation of the metal dispersion.

The average metal particle size was calculated from the metal dispersion using the following equation [24]:(1)$$d=6\frac{v}{aD_{Me}},$$
where *v* is the volume occupied by a single metal atom in the bulk of metal and a is the average surface area occupied by one metal atom. The metal dispersions, the *v* and *a* parameters, and the mean particle sizes are listed in Table 1. Consistent with the XRPD results, the calculated average particle size of Pd, Pt, and Ni was between 2 and 7 nm, which was significantly smaller than that of Ru (37.9 nm).

The N_2_ adsorption/desorption isotherms and Barrett–Joyner–Halenda (BJH) pore size distributions for the MgO–Al_2_O_3_ mixed oxide catalyst samples, calcined at different temperatures, are shown in Figure 2. Very similar isotherms were obtained for all the samples, characteristic for mesoporous materials of high specific surface area (SSA = 195–220 m^2^ g^−1^) and relatively narrow pore size distribution (1–3 nm). These results suggest that the structure of the MgO–Al_2_O_3_ mixed oxide catalysts did not change significantly upon calcination in the temperature range of 450–600 °C.

### 2.2. Acid-Base Properties

The CO_2_–TPD peak of the MgO–Al_2_O_3_ samples (not shown) could be resolved into three component peaks centered around 100, 160, and 280 °C. These peaks were attributed to weak (<100 °C), medium-strong (100–220 °C), and strong (>220 °C) base sites [13,15,22]. The concentration of the medium-strong and strong base sites in the function of the calcination temperature was obtained by integrating the corresponding peak (Figure 3A). The concentration of acid sites with different strengths was determined similarly by measuring NH_3_–TPD curves (not shown), which could also be resolved into three peaks centered at about 100, 190, and 300 °C. The peaks were attributed to weak (<100 °C), medium-strong (100–220 °C), and strong (>220 °C) acid sites [22]. The concentration of the medium-strong and strong acid sites in the function of the calcination temperature is shown in Figure 3B. Results show that the concentration of medium-strong and strong base sites decreases together at calcination temperatures above about 500 °C. In contrast, the concentration of the medium and strong acid sites changes antibatically above the same temperature.

To determine the types of acid sites present in mixed oxide catalysts, the FTIR spectrum of pyridine adsorbed on a MgO–Al_2_O_3_(550) sample was measured (Figure 4). The pair of bands at 1605 and 1445 cm^−1^ can be clearly assigned to pyridine coordinately bound to Lewis acid sites [26], whereas the absorption band of pyridinium ions characteristic for Brønsted acid sites (expected around 1540 cm^−1^) is completely missing from the spectrum (Figure 4, top spectrum). Thus, the catalyst has only Lewis-type acid sites. The intensity of the bands considerably decreased upon evacuation at ≥200 °C, substantiating that the MgO–Al_2_O_3_(550) sample contains mostly medium-strong Lewis acid sites.

### 2.3. Catalytic Properties

Results of the catalytic experiments over the MgO–Al_2_O_3_ mixed oxide catalysts calcined at 450, 500, 550, and 600 °C before the catalytic run, are shown in Figure 5A–D. Ethanol conversion (solid lines) and the composition of the organic product mixture in the function of the reaction temperature are presented (selectivity data are not shown at very low, <1%, conversion levels due to the uncertainty of the calculation). Ethanol was converted mainly to liquid products, whereas only a negligible amount of gaseous product was formed even at the highest reaction temperature (<0.5% CO and CH_4_ at 350 °C), indicating that decarbonylation/decarboxylation side reactions virtually do not take place in the absence of metallic active sites. The ethanol conversion is higher at higher reaction temperatures; however, the butanol selectivity is significantly lower mainly due to the formation of other oxygenates (mostly diethyl ether and a small amount of acetaldehyde, butanal, and butyl ethyl ether). The MgO–Al_2_O_3_(550) mixed oxide catalyst showed the highest butanol selectivity at moderate conversion levels (Figure 5C). Lower calcination temperatures (450 and 500 °C) resulted in lower ethanol conversions and/or more unsaturated alcohols (mainly crotyl alcohol and hexenol isomers) (Figure 5A,B), whereas applying a higher calcination temperature (600 °C) led to a higher formation of other oxygenates (mainly diethyl ether) and therefore lower butanol selectivity (Figure 5D). Based on these results, the MgO–Al_2_O_3_(550) mixed oxide was used as a support for the preparation of metal-containing catalysts.

The results of catalytic ethanol coupling over the metal-containing MgO–Al_2_O_3_ catalysts in inert (He) and reducing (H_2_) atmosphere are shown in Figure 6A–H, respectively. Introduction of Pd, Pt, and Ni active metals significantly improved the activity of the MgO–Al_2_O_3_ mixed oxide catalyst in both He and H_2_ carrier gas in the whole temperature range, whereas Ru was a less effective promoter especially in inert atmosphere (cf. Figure 5 and Figure 6). However, the ethanol conversions and product selectivities show some distinct differences in He and H_2_. In the presence of hydrogen, conversion of ethanol was somewhat lower at the lowest reaction temperature (200 °C) than in inert atmosphere (cf. Figure 6A–H), whereas the opposite trend was observed at higher temperatures (>200 °C). The effect of the active metal on ethanol conversion and alcohol selectivity (mainly butanol and some hexanols) was most beneficial at 250 °C, where the Pd/MgO–Al_2_O_3_ catalyst showed the most favorable catalytic properties. The alcohol selectivity together with the liquid yield considerably decreased in the high temperature range (≥300 °C) due to acceleration of the undesired side reactions over the metallic sites, especially on the Ru- and Ni-containing catalysts, leading to the formation of alkanes (mainly methane) and nonalcohol oxygenates, such as ethers, esters, aldehydes, and ketones. The selectivity loss due to the formation of latter oxygenates was considerably lower in H_2_ up to 300 °C than in He, suggesting that hydrogen can suppress reaction routes, leading to the formation of other oxygenates. At the applied highest reaction temperature (350 °C), the deoxygenation reactions (via decarbonylation and decarboxylation) were significantly accelerated, resulting in low liquid yields due to the excessive formation of gas-phase products, such as, CH_4_, CO, and CO_2_.

These results suggest that at low reaction temperatures (≤200 °C), the presence of an active metal and hydrogen may adversely affect the reaction, whereas at high temperatures (≥300 °C) the undesired hydrodeoxygenation reactions are accelerated, leading to the formation of gas-phase products (CH_4_, CO, and CO_2_) and, consequently, to low liquid yields.

### 2.4. Surface Species on MgO–Al_2_O_3_ and Pd/MgO–Al_2_O_3_ Mixed Oxide Catalysts

Formation of surface species on the surface of MgO–Al_2_O_3_(550) and Pd/MgO–Al_2_O_3_ catalysts in contact with EtOH/He or EtOH/H_2_ reactant gas mixture and their transformation products were studied using quasi-operando DRIFT spectroscopy. The experiments were carried out in transient mode at continuously increasing reaction temperature from 30 to 350 °C.

The results obtained for the MgO–Al_2_O_3_(550) catalyst in contact with an EtOH/He reactant mixture are presented in Figure 7. Adsorption of EtOH on the activated catalyst sample at room temperature resulted in the appearance of several characteristic absorption bands (Figure 7, bottom spectra) due to the formation of surface species, which was accompanied by the development of a negative absorption band at 3740 cm^−1^ in the range of the ν_OH_ stretching vibrations (see Appendix A), suggesting that surface OH-groups are involved in adsorption interaction. The adsorbed species gave main characteristic absorption bands at 2975 and 2928 cm^−1^ in the frequency range of the ν_CH_ stretching vibrations (Figure 7A, bottom spectrum), which can be assigned to the asymmetric stretching vibration of the –CH_3_ (ν_as_[CH_3_]) and –CH_2_– (ν_as_[CH_2_]) groups, respectively [27]. A third band at 2871 cm^−1^ can be attributed to the symmetric stretching vibration of the –CH_3_ groups (ν_s_[CH_3_]), probably overlapping with the band of symmetric –CH_2_– vibration (a sharp band at 2901 cm^−1^ is also discernible due to the minor contribution of gas-phase ethanol). The bands appearing below about 1700 cm^−1^ stem from adsorption of ethanol and/or its transformation products on the catalyst surface (Figure 7B). The relatively low-intensity band at 1278 cm^−1^ is assigned to the δ_OH_ deformation vibration of molecularly adsorbed ethanol [28,29,30]. At elevated temperatures, the intensity of this band decreases simultaneously with that of the bands at 1482 and 1101 cm^−1^; therefore, we assign these latter bands to the corresponding scissoring vibration of the methylene group (β_s_[CH_2_]) and the ν[C-O] vibration of molecularly adsorbed ethanol. The bands at 1165 and 1070 cm^−1^ indicate the formation of ethoxy species (both bands belong to ν[C-C-O] vibrations), most probably monodentate ethoxy species [30,31,32]. The appearance of an additional band at 1135 cm^−1^ suggests the formation of a second type of ethoxy species, which probably stems from ethoxy coordinated to Mg sites in MgO [33]. Note that the XRPD measurements confirmed the presence of MgO phase in the MgO–Al_2_O_3_ mixed oxide catalyst (*vide supra*). The bands at 1460, 1450, and 1386 cm^−1^ can be assigned to the β_s_[CH_2_], δ_as_[CH_3_], and δ_s_[CH_3_] vibrations of different adsorbed species [29,32,34]. A low-intensity band at 1361 cm^−1^ is also discernible at lower temperatures (<200 °C) in the spectra (Figure 7B, bottom spectra), which was formerly assigned to the δ[CH] vibration of adsorbed aldehyde [18,31,35]; the corresponding ν[CH] vibration appears at around 2725 cm^−1^ (Figure 7A).

The bands assigned to different ethoxy species decrease in intensity at increasing reaction temperatures, while a new pair of bands at 1585 and 1440 cm^−1^ develops and continuously gains intensity up to 350 °C (Figure 7B). These latter bands can be attributed to the ν_as_[O-C-O] and ν_s_[O-C-O] vibration of carboxyl species formed in a surface reaction from surface alkoxy (e.g., ethoxy) species, most probably via an aldehyde intermediate [18,29,30,31,32,34]. A second type of carboxylate species giving a pair of bands at about 1601 and 1410 cm^−1^ were also formed at temperatures > 200 °C. Based on the separation of the asymmetric and symmetric vibrations, the pair of bands at 1585 and 1440 cm^−1^ (Δν = 145 cm^−1^) and 1601 and 1410 cm^−1^ (Δν = 191 cm^−1^) can be attributed to chelating bidentate and bridging bidentate carboxylate species, respectively [35,36].

It is important to note that the intensities of the ν_as_[CH_3_] band at 2975 cm^−1^ and the ν_as_[CH_2_] band at 2928 cm^−1^ did not follow the same trend of temperature dependance (Figure 7A and Figure 8A). The intensity of the latter band continuously increased up to about 300 °C (Figure 8A), indicating the formation of surface-bound species having a higher number of –CH_2_– groups in the chain, that is, the formation of coupling products.

The DRIFTS results obtained for the Pd/MgO–Al_2_O_3_ catalyst in contact with EtOH/He or EtOH/H_2_ reactant gas mixture are presented in Figure 9 and Figure 10, respectively.

The DRIFT spectra obtained for the Pd/MgO–Al_2_O_3_ catalyst in contact with the EtOH/He reactant gas mixture show close resemblance to those obtained for the metal-free catalyst under the same conditions (cf. Figure 7 and Figure 9). However, some difference in the development of the absorption bands in the function of the reaction temperature is clearly visible. The intensity of the ν_as_[CH_2_] band at 2928 cm^−1^ increases somewhat steeper in the presence of Pd (cf. Figure 8A,B), suggesting that at comparable temperatures the formation of coupling products, having more –CH_2_– groups in the chain, is somewhat faster.

Similar spectra were obtained for the Pd/MgO–Al_2_O_3_ catalyst in contact with EtOH/H_2_ than with the EtOH/He reactant gas mixture (cf. Figure 9 and Figure 10) but with some distinct differences. The intensity of the ν_as_[CH_2_] band at 2928 cm^−1^ did not change significantly at temperatures below about 200 °C in hydrogen; however, it increased much steeper in the temperature range of 200–250 °C than it did in helium (Figure 8B,C), indicating that Pd and H_2_ accelerated the formation of coupling products. The beneficial effect appears to prevail in this relatively narrow temperature range, since higher temperatures (≥280 °C) also accelerated the rate of the hydrodeoxygenation reactions, as indicated by the bands of gas-phase CH_4_ (Figure 10A) and CO (see Appendix A).

It should be noted that the carboxylate species (bands at 1585/1440 and 1601/1410 cm^−1^) develop at elevated temperatures, but over about 300 °C, the intensities of their bands are considerably lower for the Pd/MgO–Al_2_O_3_ catalyst (Figure 9B and Figure 10B) than for the MgO–Al_2_O_3_ catalyst (Figure 7B). These results suggest that the steady-state surface concentration of the carboxylate species is lower in the presence than in the absence of Pd.

## 3. Discussion

The XRPD measurements revealed that the MgO–Al_2_O_3_ mixed oxide catalysts obtained by calcination of hydrotalcite (Mg/Al = 2) contains aluminum oxide finely dispersed in the MgO phase and possibly forming Mg–O–Al linkages in the mixed oxide.

The calcination temperature hardly affected the specific surface area and porosity of the catalysts but did influence the acid-base properties. The catalysts contain both basic sites and Lewis acid sites of different strengths (Figure 3). It was confirmed that calcination results in the formation of Lewis base–Lewis acid pair sites O^2−^–M^n+^ containing catalytically active medium-strong and strong base O^2−^ ions [37]. Strong base—moderate acid pairs (O^2−^–M^n+^) are believed to play an important role in the alcohol dehydrogenation step, in which the O atom is coordinated to the metal cation, and the H atom of the -OH group is abstracted by the strong base oxygen anion [4,5]. Aldol condensation requires enolate formation from the aldehyde intermediate, which was suggested to take place on strong base—weak or moderate-strength acid pairs (O^2−^–M^n+^): the O atom of the aldehyde coordinates to the metal cation, while the strong base oxygen anion abstracts a proton from the α-carbon atom. In the acid-base pair sites, Mg^2+^ ions and Al^3+^ ions are considered as weaker and stronger Lewis acid, respectively [4]. Incorporation of Al into MgO increases the quantity of appropriate strength acid-base pairs for the coupling reaction, in which the stronger Lewis acid Al was suggested to stabilize the adsorbed intermediate [4]. However, the excessive incorporation of Al results in high density of strong Lewis acid sites catalyzing undesired dehydration reactions. It was also shown that activation at high temperatures (≥600 °C) creates some free Al^3+^ sites on the catalyst surface, which are strong acid Lewis sites [37].

Our MgO–Al_2_O_3_ catalysts showed similar activity and butanol selectivity in ethanol coupling reaction under comparable reaction conditions than those published in the literature; see, for example, Refs. [2,3,4,16,22]. The present study shows that well-balanced acid-base and favorable catalytic properties can be achieved by calcination of the catalyst at 550 °C, generating strong base sites and medium-strength Lewis acid sites. At lower calcination temperature (500 °C), a significant amount of crotyl alcohol was detected in the organic liquid product, suggesting that the closing hydrogenation step of the unsaturated alcohol to butanol is less efficient on the catalyst. However, it is difficult to find a clear correlation between the observed selectivities and the acid-base properties, since the concentrations of medium-strong Lewis acid sites and strong base sites are not significantly different in the MgO–Al_2_O_3_(550) and MgO–Al_2_O_3_(500) samples. Note, however, that these data show the distribution of the acid and base sites separately but provide no information about the quantity of acid-base pairs required for the different reaction steps of the overall coupling reaction. Calcination at 550 °C seems to provide an optimum distribution of these pair sites. If high calcination temperature (≥600 °C) was applied, then at the expense of the concentration of medium-strength Lewis acid sites, strong Lewis acid sites appeared in increased concentration. These latter sites can catalyze undesired side reactions, mainly dehydration, leading to accelerated diethyl ether formation.

The MgO–Al_2_O_3_(550) mixed oxide catalyst was further modified by introducing an active metal component to promote the hydrogenation–dehydrogenation reaction (Figure 1). Pd, Pt, and Ni were present as finely dispersed metal particles (2–7 nm), whereas relatively large (~38 nm) Ru particles were formed on the catalyst surface. The Pd/MgO–Al_2_O_3_ catalyst, showing the most favorable catalytic properties, was studied in more detail.

Below about 300 °C reaction temperature, the presence of Pd significantly increased the catalytic activity without adversely affecting alcohol selectivity, which can be explained by the promotional effect of the metal in the alcohol dehydrogenation (initiation step, Figure 1) and the transfer hydrogenation of the reaction intermediates with ethanol via the MPV mechanism (closing steps, Figure 1) [4,6,12]. Interestingly, the use of H_2_ instead of inert He gas lowered the ethanol conversion at the lowest reaction temperature (200 °C), whereas significantly higher conversion was achieved in H_2_ at 250 °C than in He (Figure 6). These results indicate that the participation of H_2_ in H-transfer reactions is most probably affected by the temperature-dependent hydrogenation–dehydrogenation equilibrium. At low reaction temperatures (≤200 °C), where the equilibrium is more shifted towards hydrogenation [2,12,38], the presence of hydrogen (activated on Pd sites) hinders the initial dehydrogenation step and thereby the formation of acetaldehyde intermediate. At somewhat higher temperature (250 °C), the equilibrium starts to shift towards dehydrogenation. At this temperature, both dehydrogenation and hydrogenation are possible. Consequently, the presence of Pd and H_2_ significantly increased the ethanol conversion at around 250 °C (Figure 6). At higher reaction temperatures (≥300 °C), however, the undesired decarboxylation and/or decarbonylation side reactions were significantly accelerated, resulting in an extensive drop of butanol selectivity and a pronounced formation of gaseous products.

Results of the DRIFT spectroscopic investigations confirmed that adsorption of ethanol results in the formation of ethoxy species as the most abundant surface species. The negative absorption band observed in the ν_OH_ region suggests that ethanol adsorbed reactively with the involvement of surface OH groups, resulting in ethoxy species and water [4,30]. At elevated reaction temperatures, two kinds of surface reactions could be clearly distinguished: coupling reaction, indicated by the increasing intensity of the ν_as_[CH_2_] band (Figure 8), and formation of carboxylate species (Figure 7, Figure 9 and Figure 10). The coupling reaction proceeded already below 200 °C in the presence of Pd (cf. Figure 8A,B), suggesting that the active metal promoted the initiation step (dehydrogenation) and probably also the closing hydrogenation steps by activating C–H bonds in the ethanol reactant. Interestingly, it was found that if hydrogen is also present in the reaction system, the coupling reaction is accelerated only over 200 °C (Figure 8C). It follows that hydrogen (activated on Pd) must hinder the dehydrogenation of ethanol in the low-temperature range, where hydrogenation prevails over dehydrogenation. This result is in full agreement with the catalytic results. The coupling reaction was accompanied by the formation of thermally stable carboxylate species; however, their accumulation starts at higher reaction temperatures, and their surface concentration appears to be considerably lower over about 300 °C on the Pd/MgO–Al_2_O_3_ catalyst (Figure 9B and Figure 10B) than over the MgO–Al_2_O_3_ catalyst (Figure 7B). The carboxylate formation from ethoxy species was shown to proceed via aldehyde intermediate species [18,29,30,31,32,34]. It follows that parallel to the coupling reaction, carboxylate formation also consumes the aldehyde intermediate formed in the initiation step (Figure 1). Therefore, it is rational to think that the closing hydrogenation steps leading to saturated alcohol products (Figure 1) were accelerated in the presence of Pd, which in turn resulted in a lower carboxylate coverage on the catalyst surface. The hydrogenation proceeds via hydride transfer from an ethanol molecule in inert atmosphere, but in the presence of H_2_, hydrogenation with activated hydrogen also contributes to the acceleration of these closing reaction steps.

## 4. Materials and Methods

### 4.1. Catalyst Preparation

The hydrotalcite precursor of the MgO–Al_2_O_3_ mixed oxide catalyst with a Mg/Al atomic ratio of 2 was prepared by coprecipitation. An aqueous solution containing 102.4 g of Mg(NO_3_)_2_∙6H_2_O and 75.0 g of Al(NO_3_)_3_∙9H_2_O in 400 mL distillated water was added dropwise to 150 mL NaOH solution (pH 11.5) under vigorous stirring at room temperature. In order to keep the pH value close to 10, a solution of 2M NaOH was added dropwise to the reaction mixture simultaneously. The obtained suspension was stirred at 60 °C for 1 h and then allowed to cool to room temperature and aged overnight. The precipitate was separated by centrifugation and washed with distilled water in several consecutive steps until the pH of the washing water was below 9. The hydrotalcite sample was then placed into an oven and slowly dried at 70 °C for 2 days. The MgO–Al_2_O_3_ mixed oxide catalyst samples were obtained by calcining the hydrotalcite precursor at 450, 500, 550, and 600 °C. The obtained MgO–Al_2_O_3_ catalysts are referred to as MgO–Al_2_O_3_(450), MgO–Al_2_O_3_(500), MgO–Al_2_O_3_(550), and MgO–Al_2_O_3_(600), where the number in parentheses indicates the calcination temperature of the hydrotalcite precursor. Based on its favorable acid-base and catalytic properties (*vide supra*), the MgO–Al_2_O_3_(550) sample was used to prepare the derivatives containing Pd, Pt, Ru, and Ni. The MgO–Al_2_O_3_(550) sample was impregnated with an aqueous solution containing a calculated amount of Pd(NO_3_)_2_∙4NH_3_, Pt(NH_3_)_4_(OH)_2_·xH_2_O, RuCl_3_, or Ni(CH_3_COO)_2_∙4H_2_O metal precursor compound, respectively. The impregnated samples were calcined in air at 550 °C to decompose the precursor compound to obtain the metal-containing mixed oxide catalysts. The obtained catalysts containing 1 wt% of Pd, Pt, or Ru are referred to as 1%Pd/MgO–Al_2_O_3_, 1%Pt/MgO–Al_2_O_3_, and 1%Ru/MgO–Al_2_O_3_, whereas the catalyst containing 5 wt% of Ni is referred to as 5%Ni/MgO–Al_2_O_3_.

### 4.2. Catalysts Characterization

#### 4.2.1. X-ray Powder Diffraction Measurements

The X-ray powder diffraction (XRPD) measurements were carried out by using a Philips PW 1810/3710 diffractometer equipped with a graphite monochromator (CuK_α_ radiation, λ = 1.5418 Å) and a type HT1200 Anton Paar high-temperature chamber (Anton Paar, Graz, Austria). The XRD patterns were measured in steps of 0.02° 2*Θ* with a scan time of 5 s in each step, whereas the X-ray tube was set at 40 kV and 35 mA.

#### 4.2.2. Nitrogen Adsorption Isotherms

The specific surface area (SSA) of the MgO–Al_2_O_3_ catalyst samples previously calcined at temperatures between 450 and 600 °C was determined by measuring the adsorption isotherms of the dehydrated (pretreated at 250 °C and 10^−6^ mbar for 4 h) samples at −196 °C using an automatic, volumetric adsorption analyser (The “Surfer,” Thermo-Fisher Scientific, Waltham, MA, USA). The SSA of the samples was calculated from the N_2_ adsorption isotherms using the Brunauer–Emmett–Teller (BET) method, whereas the pore size distributions were determined by using the Barett–Joyner–Halenda (BJH) method.

#### 4.2.3. CO Pulse Chemisorption

Metal dispersion in the reduced catalysts was determined using the CO pulse chemisorption method. About 100 mg of the catalyst sample was placed into a U-shaped quartz microreactor tube and reduced in situ in a 30 cm^3^·min^−1^ H_2_–flow at 500 °C for 1 h. The sample was then purged with He flow (20 cm^3^ min^−1^) at 500 °C for 30 min and cooled to room temperature in He flow. Carbon monoxide pulses of 10 μL volume were injected sequentially in 3 min intervals into the He flow passing through the catalyst bed, while the reactor effluent was continuously monitored using a thermal conductivity detector (TCD). The signal of the TCD was processed using data acquisition software (DATAQ Instrument Hardware Manager v1.61). The injection of CO pulses was repeated until the chemisorption sites were saturated. The molar amount of chemisorbed CO per gram of the catalyst sample was calculated from the integrated areas of the TCD signals. The molar amount of chemisorbed CO was taken as being equivalent with the molar amount of surface metal atoms. The dispersion of the metal (*D*_Me_) was obtained as the ratio of the number of surface metal atoms and the total number of metal atoms in the catalyst.

#### 4.2.4. Temperature-Programmed Desorption of CO_2_ and NH_3_

The basicity and acidity of the MgO–Al_2_O_3_ catalyst samples were characterized by temperature-programed desorption of carbon dioxide (CO_2_-TPD) and ammonia (NH_3_-TPD). About 150 mg of the catalyst sample was placed into a U-shaped quartz microreactor and activated in O_2_ flow (30 cm^3^ min^−1^) at 450, 500, 550, or 600 °C for 1 h. After activation, the sample was flushed with N_2_ flow (30 cm^3^ min^−1^) for 15 min at the activation temperature, then evacuated (10^−4^ mbar) for 30 min at the same temperature and cooled to room temperature. The sample was contacted with 100 mbar of CO_2_ or NH_3_ gas, then the gas-phase CO_2_ or NH_3_ was removed from the reactor by evacuation for 15 min. The temperature-programmed desorption was carried out in He flow (20 cm^3^ min^−1^) while the reactor was heated at a rate of 10 °C min^−1^ to 700 °C. The CO_2_ or NH_3_ concentration in the outgoing gas flow was monitored by TCD. The amount of adsorbed CO_2_ or NH_3_ was determined by calculating the area under the corresponding TPD curve and using a previously determined calibration factor.

#### 4.2.5. Pyridine Adsorption

The presence of Brønsted and/or Lewis acid sites in the MgO–Al_2_O_3_ catalyst samples was confirmed by determining the Fourier-transform infrared (FT-IR) spectra of adsorbed pyridine using a Nicolet Impact Type 400 spectrometer (Thermo Scientific, Waltham, MA, USA). Spectra were collected in transmission mode by using a self-supported wafer of the catalyst with a thickness of about 15 mg cm^−2^ and averaging 512 scans at a nominal resolution of 2 cm^−1^. The design of the used IR cell allowed transferring of the sample from a built-on furnace, used for sample pretreatment at elevated temperatures, to between KBr windows to record IR spectrum. All spectra were collected at room temperature. Before pyridine adsorption, the sample was pretreated in situ in the cell under high vacuum (10^−6^ mbar) for 1 h at 450 °C. The spectrum of the activated catalyst wafer was recorded and used as a background spectrum. The sample was contacted with 5 mbar of pyridine vapor at 200 °C for 30 min and then cooled to 100 °C in the presence of pyridine vapor and was finally evacuated for 30 min to remove pyridine vapor from the cell. A sample spectrum was collected, and then the sample was evacuated at 200, 300, and 400 °C for 30 min in sequential steps. A new sample spectrum was collected after each evacuation step. The background spectrum collected before pyridine adsorption was subtracted from each sample spectra to obtain the spectra of the species obtained from pyridine.

### 4.3. Catalytic Tests

The catalytic ethanol coupling reaction was carried out in a high-pressure, fixed-bed, flow-through reactor system. The catalyst powder was pressed into wafers and then crushed and sieved to obtain the 0.315–0.630 mm sieve fraction of the particulate catalyst sample. Two grams of the catalyst were placed into the catalytic reactor (I.D.: 8 mm). Prior to the catalytic measurement, the catalyst sample was activated in situ in the reactor in H_2_ flow (50 cm^3^/min) at atmospheric pressure at 500 °C for 2 h (except the MgO–Al_2_O_3_(450) sample, which was activated at 450 °C). The reaction was carried out at 21 bar total pressure using H_2_ or He carrier gas (90 cm^3^ min^−1^) at 200, 250, 300, and 350 °C reaction temperatures and at weight hourly space velocity (WHSV) of $1\;g_{EtOH}\cdot g_{cat.}^{-1}\cdot h^{-1}$. The gas and liquid feed were controlled by a mass flow controller (Brooks, Hatfield, PA, USA) and a high-precision HPLC pump (Gilson, Madison, WI, USA), respectively, whereas the pressure of the catalytic system was set by an electronic back-pressure regulator (Brooks) placed downstream of the catalytic reactor. The ethanol feeding was continued for 1 h to reach the steady state of the catalytic system. The liquid product collected in the second hour of the reaction and in the following hours was sampled and analyzed. The gaseous effluent was also analyzed in each hour. The liquid sample collected in a high-pressure separator was first transferred to a second separator, and the gas phase over the liquid was expanded to atmospheric pressure using a large-volume sampling syringe. The gas volume collected in the syringe retains those volatile compounds, which are liquid under pressure at 21 bar but are gaseous at atmospheric pressure. Both the expanded gas phase and the effluent gas leaving the catalytic system via the back-pressure regulator were analyzed to account for all the products formed in the reaction. Liquid samples were analyzed by a gas chromatograph (GC) equipped with a flame ionization detector (FID) using a ZBWAXplus (L 30.0 m × ID 0.32 mm × df 0.25 μm) capillary column, and the gas products were analyzed by GC-TCD-FID using a ShimCarbon ST (L 2.0 m × ID 1/8 in. × OD 2.0 mm) column.

### 4.4. Quasi-Operando DRIFT Spectroscopic Investigations

Surface species obtained from the adsorption and transformation of ethanol on the catalyst were studied by Diffuse Reflectance Infrared Fourier Transform (DRIFT) spectroscopy. A Nicolet iS10 spectrometer (Thermo Scientific) was equipped with a flow-through reactor cell and a DiffuseIR mirror system (PIKE Technologies, Fitchburg, WI, USA). The finely powdered catalyst sample (about 20 mg) was placed into the sample holder of the reactor cell. The cell design allows gas flow (carrier gas or gas-phase reactant mixture) to pass through the catalyst bed with a controlled flow rate. The catalyst was pretreated in a 30 cm^3^ min^−1^ H_2_ flow at 450 °C for 1 h before the experiment and then was cooled to 30 °C in H_2_ or He flow depending on the desired carrier gas during the experiment. The adsorption/reaction of ethanol was initiated by switching the H_2_ or He flow to a gas saturator filled with ethanol (EtOH) and kept at 0 °C in an ice bath. The saturated gas passing through the catalyst bed contained 1.58 vol% ethanol reactant. The catalyst was contacted with the EtOH/H_2_ or EtOH/He mixture for 30 min at 30 °C, and then the reaction temperature was raised at a rate of 2 °C min^−1^ up to 350 °C. Spectra were taken 5 min (10 °C increments) between 30 and 350 °C. The obtained sample spectra were corrected with spectra of the catalyst in H_2_ or He flow collected at about the same temperatures. The resulting difference spectra show the bands of surface species and gas-phase molecules above the catalyst. However, the contribution of the gas-phase spectrum was proved to be minor under the applied conditions. Therefore, the obtained difference spectra practically reflect surface species formed (positive bands) or consumed (negative bands) during the adsorption and/or reaction of ethanol on the catalyst surface. Note that the catalytic system was in transient state due to the continuously increasing reaction temperature.

## 5. Conclusions

Catalytic coupling reaction of ethanol to 1-butanol was studied using MgO–Al_2_O_3_ mixed oxide catalysts prepared from hydrotalcite precursor by calcination at 450–600 °C. Relationships between acid-base and catalytic properties and the effect of active metal promoters (Pd, Pt, Ni, Ru) on the hydrogen transfer reaction steps were investigated. The highest butanol selectivity and yield was achieved on the MgO–Al_2_O_3_ catalyst calcined at 550 °C, containing both strong base and medium-strong Lewis acid sites in high concentration. The catalytic activity was improved by introducing a hydrogenating–dehydrogenating active metal component. The Pd containing MgO–Al_2_O_3_ catalyst showed the most favorable catalytic properties up to about 300 °C; however, butanol selectivity and liquid yield significantly decreased at higher temperatures (≥300 °C) due to the acceleration of undesired side reactions. Below about 200 °C, the presence of H_2_ adversely affected the initiation reaction step, which would be the formation of aldehyde intermediate, because the dehydrogenation–hydrogenation equilibrium was shifted more towards hydrogenation at this temperature. The most beneficial effect of Pd and H_2_ was observed in the relatively narrow temperature range between about 200 and 250 °C, where both the initial dehydrogenation and the closing hydrogenation steps were accelerated, but the rate of the undesired side reactions remained reasonably low.

## Data Availability

Not applicable.

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
