# Peer review of "Ethanol Coupling Reactions over MgO–Al_2_O_3_ Mixed Oxide-Based Catalysts for Producing Biofuel Additives"

_molecules, 2023, doi:10.3390/molecules28093788_

Round 1
Reviewer 1 Report
I will appreciate if authors address the following questions:
-Pretreatment temperatures, as in Figure 3 or Section 2.3, correspond to calcination temperatures?
-Catalysts XRD patterns in Figure 1 belong to MgO and do not indicate Mg—O—Al linkages; that is a deduction by the authors.
-Which are the catalysts strong basic sites? Which is their role in the reaction mechanism?
-Which are the catalyst medium-force Lewis acid sites? How do they intervene in the reaction mechanism?
-One of the authors main conclusions is: The highest butanol selectivity and yield was achieved on the MgO-Al2O3 catalyst calcined at 550 °C containing both, strong base and medium-strong Lewis acid sites. The catalyst calcined al 500 °C contains about the same amount of medium-strong Lewis acid sites and even a larger amount of strong base sites, then, why it showed a low performance?
-Experimental procedure in Section 4.3 states: The reaction was carried out at 21 bar total pressure and at 200, 250, 300 and 350 °C reaction temperatures.
-Why Figure 5 has no results for the calcined catalysts at 200 °C? The reaction did not take place at that temperature?
-At reaction temperatures higher than 300 °C undesired side reactions and the amount of byproducts become significant. However, at temperatures below 250 °C, conversion is rather low. The catalyst calcined al 550 °C was selected as the best, whereas the catalyst calcined al 600 °C showed at least the same selectivity to butanol at a 250 °C reaction temperature and about the same (low) conversion.
The text can be read without any particular difficulties, although at some points needs better use of punctuation signs and prepositions.
Author Response
Answers to the comments of REFEREE #1
- Pretreatment temperatures, as in Figure 3 or Section 2.3, correspond to calcination temperatures?
Response: Yes, both in Fig. 3 and Section 2.3 pretreatment temperatures correspond to calcination temperatures. Axis title of Fig.3 and the text in Section 2.3 was modified slightly in order to avoid misunderstanding.
- Catalysts XRD patterns in Figure 1 belong to MgO and do not indicate Mg—O—Al linkages; that is a deduction by the authors.
Response: Indeed, XRD measurement indicate that aluminum oxide is in a highly dispersed form in the MgO phase, which suggest the formation of Mg-O-Al linkages. The corresponding Result section and the Discussion section was modified in accordance with the Referee's comment.
- Which are the catalysts strong basic sites? Which is their role in the reaction mechanism?
- Which are the catalyst medium-force Lewis acid sites? How do they intervene in the reaction mechanism?
Response: The type, strength, and possible role of acid-base sites are widely discussed in the relevant scientific literature. Strong basic sites are generally attributed to low-coordination O2- anions in acid-base pair sites on the catalyst surface. Strong base – moderate acid pairs (O2––Mn+) are believed to play an important role in the alcohol dehydrogenation step, in which the O atom is coordinated to the metal cation, and the H atom of the -OH group is abstracted by the strong base oxygen anion. Aldol condensation requires enolate formation from the aldehyde intermediate, which was suggested to take place on strong base – weak or moderate strength acid pairs (O2––Mn+): the O atom of the ketone/aldehyde coordinates to the metal cation, while the strong base oxygen anion abstracts a proton from the α-carbon atom. According to our present knowledge, in the acid-base pair sites Mg2+ ions and Al3+ ions are considered as weaker and stronger Lewis acid, respectively. Incorporation of Al into MgO increase the quantity of appropriate strength acid-base pairs for the coupling reaction; the stronger Lewis acid Al is believed to stabilize the adsorbed intermediate. However, the excessive incorporation of Al and/or high calcination temperature (≥600 °C) results in high density of strong Lewis acid sites and, as a consequence, acceleration of undesired dehydration reactions.
The Discussion section was modified in order to clarify the points raised by Referee.
- One of the author’s main conclusions is: The highest butanol selectivity and yield was achieved on the MgO-Al2O3catalyst calcined at 550 °C containing both, strong base and medium-strong Lewis acid sites. The catalyst calcined al 500 °C contains about the same amount of medium-strong Lewis acid sites and even a larger amount of strong base sites, then, why it showed a low performance?
Response: Over the MgO-Al2O3(500) catalyst a significant amount of crotyl alcohol was detected in the organic liquid product, suggesting that the closing hydrogenation step of the unsaturated alcohol to butanol is less efficient on this catalyst than over the MgO-Al2O3(550) catalyst. Indeed, the concentrations of medium-strong Lewis acid sites and strong base sites are not significantly different in the two sample. Note, however, that these data show the distribution of the acid or base sites separately, but provide no information about the quantity of acid-base pairs required for the different reactions steps of the overall coupling reaction. Calcination at 550 °C seems to provide an optimum distribution of these pair sites.
The Discussion section was modified to further clarify this point.
- -Experimental procedure in Section 4.3 states: The reaction was carried out at 21 bar total pressure and at 200, 250, 300 and 350 °C reaction temperatures.
- Why Figure 5 has no results for the calcined catalysts at 200 °C? The reaction did not take place at that temperature?
Response: The ethanol conversion on the metal-free catalysts was very low (<1%) at 200 °C, which made the calculation of the product distributions very uncertain. Therefore, we do not show product selectivity data at such low conversions. Now this fact is mentioned in the Result section.
- At reaction temperatures higher than 300 °C undesired side reactions and the amount of byproducts become significant. However, at temperatures below 250 °C, conversion is rather low. The catalyst calcined at 550 °C was selected as the best, whereas the catalyst calcined al 600 °C showed at least the same selectivity to butanol at a 250 °C reaction temperature and about the same (low) conversion.
Response: The two catalysts practically showed the same product selectivity at low conversion (at 250 °C); however, the MgO-Al2O3(600) catalyst showed significantly higher dehydration activity at higher conversion levels resulting in high amount of undesired side products, mainly diethyl-ether. High dehydration activity is in line with the increased number of strong Lewis acid sites in the MgO-Al2O3(600) sample, which adversely affect the butanol selectivity. Addition of a metal component presumably do not affect the undesirably high concentration of the strong Lewis acid sites. Therefore, calcination temperature of 550 °C was chosen for the preparation of the metal containing catalysts.
- The text can be read without any particular difficulties, although at some points needs better use of punctuation signs and prepositions.
Response: The text was re-edited carefully to improve the quality of the manuscript.
Reviewer 2 Report
This manuscript has systematically investigated catalytic coupling reaction of ethanol to 1-butanol over MgO-Al2O3 mixed oxide catalysts. The influence factors of the butanol selectivity, yield and the catalytic activity have been studied in details. The format of the manuscript needs to check. In generally, the section of Materials and Methods should be between the Introduction and Results.
More details in attached file.
Author Response
Answers to the comments of REFEREE #2
- This manuscript has systematically investigated catalytic coupling reaction of ethanol to 1-butanol over MgO-Al2O3mixed oxide catalysts. The influence factors of the butanol selectivity, yield and the catalytic activity have been studied in details. The format of the manuscript needs to check. In generally, the section of Materials and Methods should be between the Introduction and Results.
Response: Although we agree with Referee that the Materials and Method section favorably precedes the Results section, the template provided by the journal requires this ordering of the sections.
- Compared with other published paper, what level will the present study achieve for ethanol to 1-butanol over MgO-Al2O3? It is better that the technology prospect is stated in the introduction.
Response: Our MgO-Al2O3 catalysts showed similar activity and butanol selectivity in ethanol coupling reaction under comparable reaction conditions than those published in the literature – see e.g. Refs. [2-4, 16, 22]. The Result section was slightly modified to indicate these comparable catalytic properties.
We would like to note that our work was fundamental research. Our goal was to better understand the relationships between the catalyst structure and catalytic activity for the MgO-Al2O3 based catalysts (mainly the role of active metal in the hydrogen transfer reaction steps was investigated), which are among the most frequently studied catalysts for ethanol coupling reaction in the scientific literature. We believe that our results presented in this paper can help to determine the directions of future catalyst design.
- Mg/Al molar ratio is defined 2 in the paper, if does other ratio have the influence on MgO-Al2O3? Which of Mg/Al molar ratio and the calcination temperature does dominate the catalytic coupling reaction?
Response: Several studies have investigated the effect of the Mg/Al ratio and the calcination temperature of the hydrotalcite precursor on the catalytic properties shown in the ethanol coupling reaction (see, e.g. the above mentioned Refs.). Both the Mg/Al ratio and the pretreatment temperature strongly influence the strength and concentration of acid-base sites in the mixed oxide catalyst. The optimal value of both parameters shall be found, where the rate of the desired reaction is high and the rates of the undesired side reactions are possibly low. We set Mg/Al ratio to 2 as this composition was considered optimal based on literature results. Since similar question was raised by Referee #1, we modified the Discussion section to give more details about acid-base properties and their role in the reaction.
- The highest butanol selectivity and yield was achieved on the MgO-Al2O3 catalyst calcined at 550℃, but the Pd containing MgO-Al2O3 catalyst showed the most favorable catalytic properties between 200 and 250℃. At last, which of two catalysts will be chosen for the catalytic coupling reaction of ethanol to 1-butanol?
Response: Our results show that introduction of Pd significantly improves the activity while preserving the selectivity of the MgO-Al2O3 mixed oxide catalyst, although in a relatively narrow temperature range (200-250 ℃). It is obvious, that the presence of an active metal can accelerate the hydrogen transfer steps and thereby the overall reaction, which strongly suggest the advantageous application of a bifunctional catalyst in the ethanol coupling reaction.
Round 2
Reviewer 1 Report
Manuscript has been improved, particularly discussion aspects and experimental information.